# Principal Component Analysis and t-Distributed Stochastic Neighbor Embedding Analysis in the Study of Quantum Approximate Optimization Algorithm Entangled and Non-Entangled Mixing Operators

**DOI:** 10.3390/e25111499

**Published:** 2023-10-30

**Authors:** Brian García Sarmina, Guo-Hua Sun, Shi-Hai Dong

**Affiliations:** 1Centro de Investigación en Computación, Instituto Politécnico Nacional, Mexico City 07738, Mexico; gsun@cic.ipn.mx; 2Research Center for Quantum Physics, Huzhou University, Huzhou 310003, China

**Keywords:** QAOA, mixing operator, entangled operator, non-entangled operator

## Abstract

In this paper, we employ PCA and t-SNE analyses to gain deeper insights into the behavior of entangled and non-entangled mixing operators within the Quantum Approximate Optimization Algorithm (QAOA) at various depths. We utilize a dataset containing optimized parameters generated for max-cut problems with cyclic and complete configurations. This dataset encompasses the resulting RZ, RX, and RY parameters for QAOA models at different depths (1L, 2L, and 3L) with or without an entanglement stage within the mixing operator. Our findings reveal distinct behaviors when processing the different parameters with PCA and t-SNE. Specifically, most of the entangled QAOA models demonstrate an enhanced capacity to preserve information in the mapping, along with a greater level of correlated information detectable by PCA and t-SNE. Analyzing the overall mapping results, a clear differentiation emerges between entangled and non-entangled models. This distinction is quantified numerically through explained variance in PCA and Kullback–Leibler divergence (post-optimization) in t-SNE. These disparities are also visually evident in the mapping data produced by both methods, with certain entangled QAOA models displaying clustering effects in both visualization techniques.

## 1. Introduction

The analysis of how Variational Quantum Algorithms (VQAs) work has been extensively studied in recent years [1,2,3,4,5]. In particular, the Quantum Approximate Optimization Algorithm (QAOA) has garnered much attention in the research community [6,7]. One of the most exciting aspects of analyzing these algorithms is understanding how they explore the problem space, what relationships exist between the rotation gates used in the circuit [6,8], and how these gates impact the overall performance of the algorithm [2,5,9]. Previous studies have attempted to shed light on these aspects [10,11] and also study the relationship between the Hamiltonian structure of a certain problem and the landscape (search space) generated [12,13]. However, as quantum hardware becomes more complex and quantum circuits become deeper, the need for a better understanding of these algorithms becomes even more critical [14,15].

In general, the analysis of QAOAs can be influenced by factors such as circuit depth and optimization strategies, which may impact the accuracy of problem results. Studies in this area often focus on the problem landscape representation, which is a critical aspect to consider when studying QAOAs from a problem resolution perspective [2,4,16,17]. However, it is important to note that there are other crucial aspects to consider when studying QAOAs, such as the extraction of information about the underlying models and their potential limitations or strengths, prior to their application to specific problems; e.g., max-sat, max-cut, Ising model, etcetera [18,19,20,21].

In this paper, our primary objective is to contribute to the existing body of knowledge by conducting an in-depth analysis of entangled and non-entangled mixing operators within the context of QAOAs. We leverage Principal Component Analysis (PCA) and t-distributed Stochastic Neighbor Embedding (t-SNE) techniques to scrutinize the parameters generated within the RZ, RX, and RY gates across various QAOA models at different depths (1L, 2L, and 3L).

Our overarching goal is to discern unique patterns of behavior that can offer valuable insights into how QAOA gate parameters behave under different scenarios; specifically, whether there are discernible differences in parameter distribution when an entanglement stage is present or absent in the mixing operator. Furthermore, we aim to present a clear and insightful visualization of these behaviors, both numerically and graphically, to enhance our understanding of the underlying dynamics of the mixing operator.

Some notable examples of visualization studies in VQAs are the works by Moussa et al. (2022) [22] and Rudolph et al. (2021) [13]. In Moussa et al. (2022), t-SNE was utilized as a preprocessing step to reveal clustering tendencies in QUBO problems and assist in determining the parameters for the QAOA. They also explored the use of supervised techniques when clusters did not adequately represent the corresponding points, leading to a more effective prediction of QAOA parameters. Rudolph et al. (2021) conducted an analysis of various visualization techniques, including PCA, applied to different VQAs. Their focus was on generating a mapping of the optimization landscape for specific problems, as well as studying aspects of parameter concentration in QAOAs, and other phenomena.

In contrast, our study focuses specifically on the representation, visualization, and information extraction from the resulting QAOA parameters, which are the RZ, RX, and RY gate values, acquired from the max-cut problem dataset. We evaluate the effectiveness of PCA and t-SNE strategies in providing comprehensive insights into the models. This evaluation encompasses both graphical representations and internal metrics derived from both methods.

## 2. Motivation and Methodology

The motivation for this work is rooted in the concepts discussed by D. Koch et al. (2020) [6], particularly in Lesson 10 referred to the QAOA. In their work, the authors raise the notion that the conventional mixing operator, which includes RX and RY gates with individual gate rotations, may prove inadequate in exploring all the feasible states within the associated Hilbert space of the system. This limitation becomes particularly prominent in scenarios involving high-dimensional spaces, where individual rotations often lead to separable states, thereby impacting the overall effectiveness of the QAOA algorithm.

In response to the challenge of limited state reachability, discussed by D. Koch et al. (2020) [6], the proposed solution involves the incorporation of an entanglement stage within the mixing operator. This modification enables access to entangled states, which constitute the majority of possible states in composite quantum systems of two or more qubits. The specific structure or properties of the entanglement stage were not detailed in their work. While there have been various studies exploring the structure of entanglement stages in quantum circuits, this remains a topic with many unanswered questions and areas for further research [23,24,25].

### 2.1. Motivation for Studying Entangled and Non-Entangled Mixing Operator

Considering the previous ideas, several questions that motivate this work arise: Is there an observable difference in the distribution of parameter values generated by the mixing operator when we introduce an entanglement stage? Can visualization techniques like PCA or t-SNE reveal visual and/or numerical disparities in the distribution of parameter values between mixing operators with and without an entanglement stage?

In our quest to address these questions, we conducted a comprehensive review of the state of the art. Our objective was to explore whether any existing research had analyzed the distribution of parameter values across a set of solutions for a specific problem, with a particular emphasis on the visualization aspect. However, our findings indicated a gap in the literature. Most research in this domain tends to analyze each individual experiment separately and often employs techniques such as heatmaps to examine the landscape of the solution space.

While we encountered some works that touched upon related aspects, such as Moussa et al. (2022) [22] and Rudolph et al. (2021) [13], these studies primarily focused on different facets of VQAs and optimization. Their specific emphases did not align with the questions that we were trying to solve in our research. Consequently, our study represents a unique contribution to the field, shedding light on the distribution of parameter values in the context of entangled and non-entangled mixing operators within the QAOA.

In the following sections, we explain the results obtained to answer these questions, where indeed the results show differences in the mapping data (both numerically and visually) using PCA and t-SNE as our visualization techniques when we encounter an entanglement stage in the mixing operator.

### 2.2. Methodology of QAOA Dataset Usage

In this paper, we utilized a dataset containing the optimized parameters acquired for the phase (RZ gates) and mixing operators (RX and RY gates). These parameters were obtained by applying the Quantum Approximate Optimization Algorithm (QAOA) in conjunction with the Stochastic Hill Climbing with Random Restarts (SHC-RR) optimization method to a series of max-cut problems. SHC-RR does not exhibit a specific tendency in its exploration of the search space, making it a more unbiased strategy suitable for data generation.

The dataset of optimized max-cut problems was created for an upcoming study, which also involves an analysis of QAOAs. In this work, we do not delve into the methodology of solving the max-cut problems using QAOAs or assess the quality of the optimized solutions provided in the dataset. Our sole focus is on utilizing the generated parameters associated with the optimized solutions for a set of max-cut problems. For comprehensive results, including the optimized parameters, please refer to reference [26].

Additionally, it is important to note that our analysis in this study does not consider the expected energy value or evaluated cost obtained from the solution of a particular experiment in a max-cut problem.

The dataset comprises the parameters obtained with QAOAs using SHC-RR to solve max-cut problems with cyclic and complete configurations, involving different numbers of nodes: 4 nodes (4n), 10 nodes (10n), and 15 nodes (15n). Each problem was simulated 100 times, where different QAOA depths were tested, including 1L, 2L, and 3L. Additionally, each QAOA model was evaluated both with and without an entanglement stage (for every depth) in the mixing operators.

The dataset generated for a particular model and problem contains 100 simulations, each representing different solution scenarios due to the inherent variability of SHC-RR in parameter distributions. Therefore, we do not consider the quality of the solution, as each simulation could yield a better or worse-optimized result. The primary focus of this research is to extract the properties of the parameters without factoring in the quality of the solution. This approach is valid because the only difference between the compared models is the presence or absence of the entanglement stage in the mixing operator. In this comparison, the depth (1L, 2L, or 3L), problem type (configuration and number of nodes), and optimization method (SHC-RR) are held constant across all compared models.

PCA and t-SNE are employed in two distinct ways to analyze the QAOA models. In individual analysis, each method is applied to a specific QAOA model depth either with or without an entanglement stage in the mixing operator, for a particular max-cut problem dataset. This yields numeric and graphical results for each method, allowing us to compare the individual outcomes.

In paired analysis, we directly compare the entangled and non-entangled QAOA models (with the same depth) using a single PCA or t-SNE model. In other words, one PCA or t-SNE model is applied to the values from both datasets of QAOA parameters, representing the entangled and non-entangled versions of the QAOA model at a specific depth.

These two approaches applied to the QAOA datasets provide us with a comprehensive understanding of how the entangled and non-entangled mixing operators behave under different conditions.

## 3. Problems to Analyze

To provide a comprehensive foundation for understanding the differences between entangled and non-entangled mixing operators in the context of the Hamiltonian and circuit structure, this section has been developed.

We commence by offering a general overview of the max-cut problem, which serves as the basis for the dataset’s content. Subsequently, we explore the fundamental structure of the phase operator, which plays a crucial role in comprehending the two distinct types of max-cut problems found in the dataset. Finally, we arrive at the crux of our investigation: the representation of both entangled and non-entangled mixing operators. This representation is of paramount importance, as it sets the stage for subsequent visualization and numerical analyses. It also plays a pivotal role in differentiating the entangled and non-entangled QAOA models, alongside the depth factor.

### 3.1. Max-Cut Problem

A maximum cut (max-cut) problem is a combinatorial optimization problem that is often used in the field of both quantum and classical computer science, and operations research. In this problem, one is given an undirected graph where each edge is associated with a weight, and the goal is to partition the vertices of the graph into two sets, called A and B, in such a way that the sum of the weights of the edges that cross between these two sets is maximized.

In Figure 1, we illustrate the process of solving a max-cut problem using a simple example. The graph in question consists of four nodes labeled from 1 to 4. To find the optimal solution, we divide the graph into two distinct groups: Group A, comprising nodes 1 and 3, and Group B, comprising nodes 2 and 4. It is important to note that, in this example and in the parameters dataset, the weighted connections between nodes are considered to be unitary, each with a value of 1. Given this assumption, the solution comes from determining the number of connections that cross between the two groups, which transition from the green-colored nodes (Group A) to the red-colored nodes (Group B). In this particular example, we observe four such connections.

When applying QAOA to address max-cut problems, the solution entails a set of values assigned to the gates of the phase and mixing operators, effectively determining the state with the highest probability of representing the optimal solution. Referring back to our previous example, this solution could manifest as either |0101〉 or |1010〉 state, each with an associated probability. In this representation, qubits in state 0 correspond to one group, while those in state 1 correspond to the other group. The aim is to maximize the probability of obtaining the correct solution state, ideally approaching a probability close to 100%, contingent on the quality of the QAOA model, which encompasses factors such as gate parameter precision and the number of parameters employed.

### 3.2. Hamiltonian and Circuit Description

Transitioning to the discussion of the Hamiltonian and circuit description, we begin by elucidating the distinctions between the non-entangled and entangled mixing operators, as depicted in the figure below.

Within the dataset, each of the QAOA models’ depth has a different number of parameters, namely, the 1L model has one set of RZ, RZ, and RY parameter values, the 2L model has two sets of RZ, RX, and RY parameter values, and the 3L model has three sets of RZ, RX, and RY parameter values. Each QAOA model is evaluated both with and without the inclusion of an entanglement stage in the mixing operator (see Figure 2).

In Figure 3, we provide a visual representation of the various levels of QAOA quantum circuit depths. Each of the 1L, 2L, and 3L depths corresponds to the number of pairs of phase and mixing operator applications in the QAOA model dataset, 1L being a pair of operators, 2L two pairs of operators, and 3L three pairs of operators. Detailed explanations of these operators will be presented in the following sections.

As a side note, each of the 10 and 15-node problem datasets contain experiments using 1L, 2L, and 3L depths. However, for the four-node problem datasets, we have only depths of 1L and 2L due to their relatively simpler nature.

The Hamiltonian configurations for dataset problems can be viewed in the following phase operators for the cyclic and complete configuration.
(1)$$U\left(H_{cyc},\gamma \right)=e^{-i\gamma H_{cyc}}=\prod_{\langle j,k\rangle }^{}e^{-i\gamma Z_{j}Z_{k}}$$

In the max-cut problem with the cyclic configuration, the representation of the phase operator can be expressed using Equation (Equation 1), where 〈j,k〉 denotes the notation for neighboring nodes. For instance, in the case of the four-node problem, this equation can be interpreted as connections between nodes 1 to 2, nodes 2 to 3, nodes 3 to 4, and nodes 4 to 1, where the final connection completes the cycle.
(2)$$U\left(H_{com},\gamma \right)=e^{-i\gamma H_{com}}=\prod_{\left\{j,k|j\neq k\right\}}^{}e^{-i\gamma Z_{j}Z_{k}}$$

For the complete configuration, the representation for the phase operator follows Equation (Equation 2). In this case, there is a connection between every pair of nodes in the graph, excluding self-connections $\left\{j,k|j\neq k\right\}$. Also, connections between nodes are not repeated, meaning that a connection from node *j* to node *k* is considered the same as a connection from node *k* to node *j*. This is due to the absence of directionality in the max-cut problem.
(3)$$U\left(H_{B},\beta_{1},\beta_{2}\right)=e^{i\beta_{1}\beta_{2}H_{B}}=\prod_{j}^{}e^{i\beta_{1}X_{j}}e^{i\beta_{2}Y_{j}},$$

The mixing operator without entanglement for both max-cut configurations is represented by Equation (Equation 3). This equation includes RX and RY rotations in the mixing operator with the associated parameters for each gate.

The entangled mixing operator includes an additional term compared to the non-entangled case. This term is generated by applying CNOT gates between each qubit (node) in the system, similar to the complete configuration. The equation representing the CNOT action is as follows:(4)$$e^{i\frac{\pi }{4}\left(I-Z\right)⊗\left(I-X\right)}=\left(\begin{array}{cc} I & 0\\ 0 & X \end{array}\right)=\left(\begin{array}{cccc} 1 & 0 & 0 & 0\\ 0 & 1 & 0 & 0\\ 0 & 0 & 0 & 1\\ 0 & 0 & 1 & 0 \end{array}\right),$$
to represent the entangled mixing operator in our notation, we use the expression $e^{iI_{j}X_{k}}$ to indicate that qubit *j* is controlling qubit *k* as the target. By adding the term $e^{iI_{j}X_{k}}$ to the previous mixing operator without entanglement, we obtain the following expression:
(5)$$U\left(H_{B},\beta_{1},\beta_{2}\right)=e^{i\beta_{1}\beta_{2}H_{B}}\;=\prod_{j}^{}e^{i\beta_{1}X_{j}}\prod_{\left\{j,k|j\neq k\right\}}^{}e^{iI_{j}X_{k}}\prod_{j}^{}e^{i\beta_{2}Y_{j}},$$
which represents the mixing operator with an entanglement stage between the RX and RY rotations. The entanglement stage generates interactions between each pair of qubits in the system, ensuring that there are no repeated interactions and no self-interactions.

The quantum circuit representation of the two types of tested mixing operators can be seen in Figure 4 and Figure 5. Figure 4 corresponds to the non-entangled mixing operator, while Figure 5 corresponds to the entangled mixing operator.

Both methods (PCA and t-SNE) are applied to each model, where the models are 1L considering one phase operator with one associated parameter γ and one mixing operator with two associated parameters $\beta_{1}$ and $\beta_{2}$; a 2L model that has two phase operators with $\gamma_{1}$ and $\gamma_{2}$ and two mixing operators (connected between each phase operator, one per operator) with $\beta_{1-1}$, $\beta_{1-2}$, and $\beta_{2-1}$ and $\beta_{2-2}$; and a 3L model that was tested for some problems (not all) with $\gamma_{1}$, $\gamma_{2}$, and $\gamma_{3}$ with the corresponding six parameters for the three mixing operators (two for each operator, as in the previous cases).

## 4. PCA and t-SNE Description

In this section, we explain how the PCA and t-SNE approach is used to analyze the properties of the QAOA dataset.

### 4.1. PCA

Principal component analysis is a method (with statistical or geometric interpretation) that aims to reduce the dimensionality of a dataset, retaining as much of the original information as possible [27,28].

PCA operates by employing specific structures known as principal components, which are designed to capture the maximum variance in the directions they are projected. Utilizing these principal components, we have the ability to transform the original data into a new coordinate system. Typically, the first two principal components are used for this transformation, enabling the data to be visualized in a more interpretable and meaningful manner [27,28].

In our study, our PCA process commenced with the calculation of the covariance between a pair of the resulting parameter values of RZ, RX, and RY gates obtained from a particular experiment within the dataset. It is important to note that there could be multiple sets of values depending on the model under consideration.
(6)$$cov\left(i,j\right)=\frac{1}{n-1}\sum_{k=1}^{n}x_{k,i}\;x_{k,j}$$

We use Equation (Equation 6) to calculate the covariance between parameters of the RZ and RX, RZ and RY, and RX and RY gates, where *n* takes a value of 100, which is the number of experiments performed for each QAOA model in a particular max-cut problem.
(7)$$\Sigma =\sum_{i}^{}\sum_{j}^{}cov\left(i,j\right)$$

Then, we calculate the eigenvalues and eigenvectors of the resulting covariance matrix (Equation (Equation 7)) for the parameters of a certain model in the dataset.

Let i=1,2,…,n, where n<N is the number of principal components and *N* is the dimension of the original data. Let $\theta =[\theta_{1},\theta_{2},\ldots ,\theta_{N}]$ be the original data vector, $P_{i}^{T}$ be the transpose of the eigenvector matrix (obtained using Σ), and $\phi =[\phi_{1},\phi_{2},\ldots ,\phi_{n}]$ be the resulting transformed vector in the principal component space. The projection using PCA can be described as follows:(8)$$\phi_{i}=P_{i}^{T}\theta_{i},$$
where Equation (Equation 8) represents the projection (via the dot product between $P_{i}^{T}$ and $\theta_{i}$) onto a principal component *i* represented by $\phi_{i}$. This process is then repeated to obtain all the principal components, where each new component is orthogonal to the previous ones.
(9)$$Var\left(\phi_{1}\right)\geq Var\left(\phi_{2}\right)\geq \cdots \geq Var\left(\phi_{i}\right)>0$$

The variance of the principal components follows the relationship described by Equation (Equation 9). This equation indicates that the variance of the principal components generally decreases as the index increases. Consequently, higher index values correspond to a reduced amount of variance information contained in the data.

### 4.2. t-SNE

The t-distributed stochastic neighbor embedding method is similar to PCA in the sense that it is used as an algorithm for data visualization and dimensionality reduction. The main difference (besides the methodology) with t-SNE is its capability to represent non-linear relationships in the data and its ability to preserve the high-dimensional structure of the original data into a lower-dimensional space [29,30].

The t-SNE algorithm creates pairwise similarities using a Gaussian kernel by measuring the distance between the points in the original dataset. Then, the algorithm generates probability distributions over pairs of points, where the probability of being similar is related to the pairwise similarity. The resulting selected objects get mapped to a similar probability distribution in a lower-dimensional space. The algorithm minimizes the difference between the two selected distributions with the objective of finding a lower-dimensional representation that preserves the original data structure [29,30].

Similar to the PCA method, our approach involves utilizing the parameters of the RZ, RX, and RY gates for algorithm development. In this instance, we compute pairwise similarities by measuring the distance between each pair of parameter gate values within the dataset.
(10)$$p_{ij}=\frac{p_{j|i}+p_{i|j}}{2n}$$

In the equations, $p_{j|i}$ and $p_{i|j}$ represent the conditional probabilities of a point *j* given point *i* and vice versa, with *n* denoting the total number of points in the dataset. Notably, for t-SNE, $p_{ii}$ and $p_{jj}$ are both zero, and $p_{ij}$ is equivalent to $p_{ji}$. Equation (Equation 10) is responsible for computing pairwise similarities within the original space. For instance, it calculates the similarity between a specific parameter of an RZ gate and the parameter of an RX gate. This calculation is repeated for each pair of parameter gate values in the dataset, specific to a given QAOA model. Then,
(11)$$q_{ij}=\frac{{\left(1+{∥y_{i}-y_{j}∥}^{2}\right)}^{-1}}{\sum_{k\neq l}^{}{\left(1+{∥y_{k}-y_{l}∥}^{2}\right)}^{-1}},$$
is applied to create the map candidates $y=y_{1},y_{2},\ldots ,y_{n}$ in the lower dimensional space. These candidates are initially set randomly, commonly using a Gaussian distribution with a small variance centered at the origin. In order to find the best mapping relations, t-SNE minimizes the Kullback–Leibler divergence, given by:(12)$$KL\left(P||Q\right)=\sum_{ij}^{}p_{ij}log\left(\frac{p_{ij}}{q_{ij}}\right)$$

## 5. Experiments and Results

The PCA method attempts to find correlations between data by analyzing the variance between points in the dataset and then reducing the original attributes into a new basis with fewer dimensions. For the experiments performed in this study, we also obtained the explained variance to identify how much variance or information is presented in each component. To compare the entangled and non-entangled models, we performed a PCA model individually and for each pair of models (with compatible dimensions) that differed only in the entanglement stage of the mixing operator.

In t-SNE, we follow a similar procedure; however, t-SNE provides us with information about the relationships between the data using a different method. Specifically, t-SNE generates dimensionality reduction by modeling the data as a pairwise probability distribution, where each distribution represents the likelihood of each data point being related to other data points. During the process of representing the data in a lower-dimensional space, the algorithm reduces the Kullback–Leibler divergence, which is based on the relative Shannon entropy. The new represented data hold as much information as possible from the original data points. In our experiments, we also obtained the Kullback–Leibler divergence (KL-Divergence) after optimization, which represents the amount of information loss in the final embedding. A low divergence value is generally considered as better, as it indicates that the low-dimensional embedding is a good representation of the high-dimensional data, while a high divergence value indicates a significant loss of information in the final embedding. In simpler terms, a lower KL-Divergence value signifies better information preservation. This means that there are more detectable correlations between the data when utilizing t-SNE.

### 5.1. PCA Applied to QAOA Dataset

In our application of PCA to each dataset, we initially performed an individual PCA analysis using the first three PCA components and recorded the corresponding explained variance. Subsequently, we compared the individual PCA projections by pairing models that had the same number of parameter gate values (same dimensions). This comparison involved using the first three PCA components of both models and examining how the combined PCA projection differed from the original individual maps. This allowed us to assess the variations between the projections.

It is important to note that the PCA individual and paired approaches were applied to all three different levels of depth to establish a fair basis for comparison between models. For the 1L depth level, which corresponds to the 3p model (three parameters), PCA is not necessary for dimensionality reduction since the number of parameters is equal to the number of PCA components that we are seeking. However, employing PCA in this case allows us to identify correlations between the parameters, indicating the relative importance of certain gates within the QAOA. For the 2L and 3L depth levels, corresponding to the 6p (six parameters) and 9p (nine parameters) models, respectively, the first three PCA components provide information about parameter correlations within the QAOA as well as dimensionality reduction.

Table 1 presents the explained variances for the individual PCA projections for the 4n cyclic configuration max-cut problem. In the case of the first 1L model, which comprises three parameters, we observed that the second row (3p entangled model) exhibited a decrease in correlation in the first PCA component compared to the non-entangled model. This decrease in correlation suggests a reduced significance of this component in terms of representation importance.

However, the second and third values increased in the 3p entangled model, indicating an increased contribution to the variance for the remaining PCA components.

For the 6p parameter models, the entangled model demonstrated higher explained variance values when comparing the first three components. Consequently, the total amount of variance contained in the components increased.

In Figure 6, we present individual PCA graphs for the 4n cyclic max-cut problem. The first two graphs of the red model (non-entangled 3p) exhibit particular line patterns for PCA 1 vs. PCA 2 and PCA 1 vs. PCA 3.

For the blue (entangled 3p) model, a similar line behavior is observed in the first graph (PCA 1 vs. PCA 2) with one more line compared to the non-entangled model. For the last two graphs, PCA 1 vs. PCA 3 and PCA 2 vs. PCA 3, a separation pattern with two groups is visible.

In the 6p parameter (2L depth) models, the green model (non-entangled) does not exhibit any recognizable pattern or cluster in the graphs. For the purple (entangled) model, PCA 1 vs. PCA 2 has three distinct cluster lines, but no recognizable patterns are observed in the rest of the planes.

In the case of using a pair PCA model in the 4n cyclic max-cut problem for the 3p and 6p, the results for the PCA components are shown in Table 2. The PCA explained values for the 3p pair model follow an intermediate trend between the entangled and non-entangled models of the separate PCA model.

For the 6p pair model, the variance of the PCA components seems to be closer to the 6p non-entangled model from the separate PCA models. Additionally, the sum of the first three components for the 6p accounts for only 60% of the information variance, which indicates that, for that type of model, it is harder to find a specific trend due to the low original information maintained in the new map.

In Figure 7, we present the pair PCA model for the 3p and 6p models. In the 3p models (red non-entangled and blue entangled), we observe that the behavior from the individual graphs is preserved. However, when we have both entangled and non-entangled models, we can see how the data of the models are projected in different areas while still following the same patterns as in the previous graphs.

In the 6p models, the previous patterns do not hold, and the distribution of projected points seems to be random in the majority of the graphs. Only the PCA 1 vs. PCA 3 graph shows some pattern with small centered line clusters for the entangled model (purple), while the non-entangled model (green) is more scattered compared to the purple data.

### 5.2. t-SNE Applied to QAOA Dataset

t-SNE was used as an additional method to identify patterns in the QAOA dataset. We aim to have multiple tools to extract information about the entanglement stage and investigate how these stages affect the overall relationships between the data.

We used different perplexity values for t-SNE analysis, which is an important parameter that determines the number of nearest neighbors used in the lower dimensional representation. We tested three different values of perplexity—3, 30 (the default value in the Sklearn package), and 99 (an extra value of 199 only for pair models)—with the goal of identifying different data behaviors at different perplexity levels. We used PCA for initialization embedding (inside t-SNE) as it provides a more globally stable solution compared to random initialization, which allows for a more precise comparison between models.

Also, as in the case of PCA, we created individual and paired t-SNE graphs for each dataset problem and for each model using 3p parameters, 6p parameters, and 9p parameters (for the 10n and 15n problems).

For the first problem dataset, at a perplexity of 3, the non-entangled models had higher (worse) KL-D values after the mapping compared to the entangled models as shown in Table 3. At a perplexity of 30, the 3p non-entangled model had a better (lower) KL-D value compared to the entangled model, and for the 6p models, the entangled model still had a better KL-D value compared to the non-entangled one.

Finally, at 99 perplexity, all models showed good KL-D values, indicating a better mapping for all the perplexity values tested, where the non-entangled and entangled models had a slight difference and the non-entangled models performed slightly better at this perplexity; however, this can be considered as negligible.

The individual t-SNE graphs for the 4n cyclic max-cut problem are shown in Figure 8. For the 3p non-entangled model (red), the most significant pattern can be observed at the 99 perplexity level, which has a linear pattern similar to the one obtained in the PCA graph for that particular model and problem dataset. For the 3p entangled model (blue), the 30 perplexity level shows a two-cluster pattern, and the 99 perplexity level shows a circular pattern with no data points in the center of the plane.

For the 6p non-entangled model (green), the 30 perplexity level has a similar distribution to the one obtained in the PCA individual graph for the same model, with a random distribution pattern. And, for the 6p entangled model (purple), the most significant pattern can be observed at the 99 perplexity level, which has an external circle with a middle line pattern.

The KL-D values for the pair models in the 4n cyclic max-cut problem are presented in Table 4. Interestingly, all the best KL-D values were obtained by the 3p models. Consistent with the individual t-SNE analysis, the best KL-D values were obtained with the highest perplexity value.

In the pair t-SNE models graphs (Figure 9) for the 4n cyclic max-cut problem dataset, we start by focusing on the 3p models non-entangled and entangled (red and blue, respectively) at 199 perplexity. The line patterns of the red model are maintained, but the blue model shows a completely different distribution, where it has a similar pattern to the red model. The entangled model data contain the red points at the center, but, at the extremes, the red model seems to contain the blue data.

For the 6p models non-entangled and entangled (green and purple, respectively), interesting results are seen at the 99 and 199 perplexity values. The purple model tends to be grouped in certain areas of the plane at 99 perplexity, while the green model has a random distribution in the plane with no particular pattern. For 199 perplexity, the patterns seen in the individual graphs are maintained, with an elliptical behavior, and, in particular, the purple model shows a line pattern at the center.

### 5.3. Results Analysis

In this subsection, we provide a comprehensive overview of our findings. We observed distinct parameter distribution patterns between entangled and non-entangled models across all datasets, whether in individual or paired analysis. Furthermore, numerical disparities were evident, as seen in the explained variance for PCA and KL-Divergence in t-SNE, highlighting the differences between the two mixing operator variants.

In the results obtained from the PCA method, the 3p models (corresponding to the 1L depth) for both cyclic and complete max-cut problems exhibit the best values for explained variance in the first three components. This is due to the fact that these models have the same number of parameters (dimensions) as the number of PCA components, resulting in no dimensionality reduction and no loss of information. This characteristic sets these models apart, and it is interesting to observe that there are differences between the components, indicating correlations between certain gate parameters within the QAOA. However, further studies are needed to determine the specific interactions between gates that are more significant, which are perceived as greater variances for certain PCA components.

From a graphical perspective, the 3p models tend to exhibit linear pattern behaviors, where the type of problem where the parameter values come from contributes to a certain consistency in the observed patterns.

In the 6p models (corresponding to 2L depth) analyzed using PCA, we observe more interesting behaviors due to the increased complexity of these models. The dimensionality reduction provided by the first three PCA components allows for a more effective application of the PCA strategy overall.

Examining the results of explained variances in the individual PCA graphs for the cyclic configuration problem datasets of 4n, 10n, and 15n (Table 1, Table A3 and Table A7), we find that the entangled models consistently yield to better (higher) values in the PCA components. This behavior is interesting as it indicates a discernible difference between the entangled and non-entangled models. It suggests that the presence of the entanglement stage in the mixing operator leads to a greater amount of information (variance) contained in the QAOA parameters, which can be detected and maintained by the PCA method.

From a graphical standpoint, focusing on the PCA 1 vs. PCA 2 plane, which contains the most relevant or informative data, as seen in Figure 6, Figure A3 and Figure A9, there are noticeable differences between the non-entangled (green) and entangled (purple) models. In the case of the non-entangled models, the distribution of mapped data appears to be random, which can be attributed to the individual rotations (gates) of the model. On the other hand, the entangled models exhibit clustering behaviors, leading to distinct visual differences in the graphs. Despite the PCA method being unaware of the fact that the processed data originate from an entangled circuit, it is capable of detecting and representing the differences in data distribution.

Also, in the case of the 6p models for complete configuration problems, the 4n, 10n, and 15n problem datasets (Table A1, Table A5 and Table A9), we observe a similar behavior as seen in the cyclic problem datasets. The entangled models consistently exhibit higher values in the PCA components compared to the non-entangled models, which could be seen in the individual variance for each PCA component (most cases) and the total amount of variance contained by the PCA model (all cases).

Examining the graphical representations (Figure A1, Figure A6 and Figure A12), most of the entangled models exhibit clustering behaviors, while the non-entangled models do not show a clear pattern or distribution. These observations further highlight the distinguishing characteristics between entangled and non-entangled models in terms of their PCA representations.

For the 9p models (3L depth), both in the cyclic and complete configurations, we once again observe higher PCA values for the entangled models regardless of the type of problem. However, it is important to note that the total amount of variance in the 9p models is relatively low. Consequently, when examining the graphical representations (Figure A4, Figure A7, Figure A10 and Figure A13), we should not draw definitive conclusions. The observed behaviors or patterns in the graphs tend to vary from one problem to another. Therefore, further analysis and investigation are needed to fully understand the implications of the PCA results for 9p (or more complex) models.

In the pair PCA models, we observed a decreasing trend in variances as the number of parameters increased, namely for the 3p, 6p, and 9p models. It is important to note that the 3p models should not be compared in the same manner as the 6p and 9p models due to the number of PCA components generated.

The purpose of the paired graphs (Figure 7, Figure A2, Figure A5, Figure A8, Figure A11 and Figure A14) was to determine if the individual behaviors could be captured within a pair PCA. This would suggest that differences between parameters in QAOA models could be detected within the same PCA. In most cases, the individual behaviors were indeed maintained in the pair graphs, supporting the notion that distinct parameter characteristics could be identified using the pair PCA approach.

For the t-SNE analysis, the results presented in Table 3, Table A11, Table A13, Table A15, Table A17 and Table A19 show a clear tendency of generating better (lower) KL-Divergence values for the entangled models, with the difference being more pronounced depending on the perplexity value.

In the individual t-SNE analysis, the best results were generally generated by the 3p models. This can be attributed to the fact that these models have only three parameters (only one set of RZ, RX, and RY gate parameters), and the t-SNE projection to the plane does not lose a significant amount of information in the process. The best KL values were reported at 99 perplexity, where all the models generated good values that were closer to zero.

For the paired t-SNE models presented in Table 4, Table A12, Table A14, Table A16, Table A18 and Table A20, the best KL values were also reported for the 3p models. In this case, the quality of the reported values decreased as the number of parameters increased for the majority of perplexity values tested.

The worst KL values were obtained at 30 perplexity. At 199 perplexity, we obtained the best KL values, similar to the individual case at 99 perplexity. Most paired models generated good KL-Divergence values, indicating a better representation in the low-dimensional space when using the t-SNE method.

In the graphical results for the individual t-SNE models presented in Figure 8, Figure A15, Figure A17, Figure A18, Figure A20, Figure A21, Figure A23, Figure A24, Figure A26 and Figure A27, we selected the graphs generated at 99 perplexity as the best representation due to the value of the KL-Divergence value.

In the 3p models, we observed similar behaviors as those observed in the PCA graphs. The non-entangled 3p model (red) consistently exhibited a three-line cluster pattern across different problems and depths of the QAOA model. On the other hand, the entangled 3p model (blue) showed varying patterns depending on the problem type and QAOA depth, often generating line clustering patterns, although they were not as well-defined as those of the non-entangled model.

For the 6p models, we observed similar patterns between the entangled (purple) and non-entangled (green) models. At 99 perplexity, most of the graphs displayed an elliptical pattern, with the entangled models sometimes exhibiting more pronounced grouping behavior in certain areas of the plane. The same behavior observed in the 6p models was also reported in the more complex 9p models, where, at the highest perplexity, both non-entangled (orange) and entangled (brown) models generated elliptical patterns, with the non-entangled models exhibiting a more evenly distributed pattern around the ellipse.

In the paired t-SNE graphical results presented in Figure 9, Figure A16, Figure A19, Figure A22, Figure A25 and Figure A28, we observed distinct patterns and behaviors. In the 3p models, at certain perplexity values, particularly higher perplexity values like 99 or 199, clear differences were observed between the non-entangled (red) and entangled (blue) models. This indicates that the paired t-SNE is capable of differentiating between different types of data within the model.

In some cases, the distributions observed in the individual graphs were maintained in the paired t-SNE plot, while, in other cases, similarities with the PCA graphs were observed. For the 6p models, significant differences were observed between the non-entangled (green) and entangled (purple) models. The non-entangled model tended to exhibit a more random distribution in the t-SNE plane across different perplexity values, while the entangled model showed a tendency to be more concentrated in certain areas. At 199 perplexity, both models recreated the elliptical behavior observed in the individual graphs, where the entangled model exhibited differences compared to the individual graphs.

Specifically, the entangled model showed more clusters around a certain distribution, while the non-entangled model continued to exhibit a more evenly distributed pattern.

For the 9p models, similar behaviors were observed as in the 6p models. The non-entangled model (orange) tended to be more evenly distributed in the t-SNE plane across different perplexity values, and a clear elliptical pattern was generated at 199 perplexity. On the other hand, the entangled model (brown) displayed a tendency to be more grouped in certain areas of the plane at different perplexity values. At 199 perplexity, the entangled models followed the elliptical pattern but appeared more compressed in certain areas of the distribution.

## 6. Conclusions

PCA and t-SNE show graphical and numerical differences between the parameter distributions of the QAOA models: the entangled models achieved greater correlations while the non-entangled models showed lower levels of correlations between parameters in the different QAOA models datasets.

The PCA method reveals differences in the amount of variance contained in the PCA components depending on the type of model dataset processed. The entangled models consistently exhibit higher variance values, either in each PCA component or in the total amount of variance.

However, the PCA method is not suitable for achieving good mapping in a low-dimensional space for the datasets investigated in this work. We observed a significant reduction in the amount of information captured by the PCA components as the number of parameters increased. The most complex model tested (9p parameters, 3L layers of depth) usually contained less than 50% of the original variance in the first three PCA components.

In some cases, the paired PCA graphs were able to retain the patterns observed in the individual PCA graphs, which is important for visually distinguishing between entangled and non-entangled models.

In general, t-SNE, whether applied to individual or paired models, outperformed the PCA method. This can be observed from the KL-Divergence values obtained at different perplexities, indicating a better representation in the low-dimensional space.

In the individual t-SNE models, we also noticed variations in the KL values between non-entangled and entangled models. Broadly speaking, the entangled models displayed superior (lower) KL values, which we attribute to the presence of the entanglement stage in the mixing operator. This stage enhances the capacity to preserve correlation information arising from the relationships between parameter gate values, something that the t-SNE models can effectively capture.

Lastly, in the paired t-SNE models, clear differences were observed between non-entangled and entangled models at various perplexity values. For 3p models, more linear pattern distributions were observed, while for 6p and 9p models, non-entangled models exhibited more random and elliptical distributions, whereas entangled models displayed a tendency to cluster while following a certain pattern depending on the dataset. These findings highlight the ability of t-SNE to visually distinguish the differences in data relationships between non-entangled and entangled models.

## 7. Future Work

For future research, it is important to conduct additional investigation into the interpretation of the observed distributions. At present, it is premature to conclude whether these specific patterns occur universally in all models with an entanglement stage, regardless of the problem. It is also unclear whether different patterns may emerge in other types of problems, indicating the presence or absence of an entanglement stage in QAOAs. Further exploration and analysis are necessary to gain a comprehensive understanding of these phenomena.

Additionally, it would be valuable to explore alternative optimization methods for QAOA dataset generation in order to compare the obtained results. This analysis would help to identify which behaviors persist across different optimization methods and which ones are influenced by the specific method employed to solve the presented problems. 

## Figures and Tables

**Figure 1 entropy-25-01499-f001:**
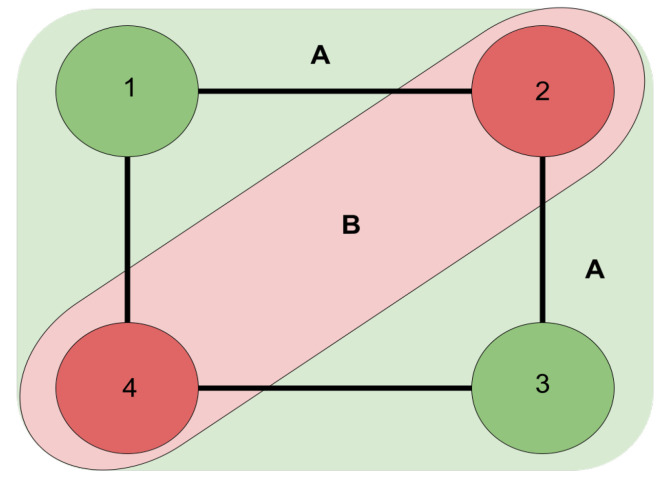
Max-cut example.

**Figure 2 entropy-25-01499-f002:**
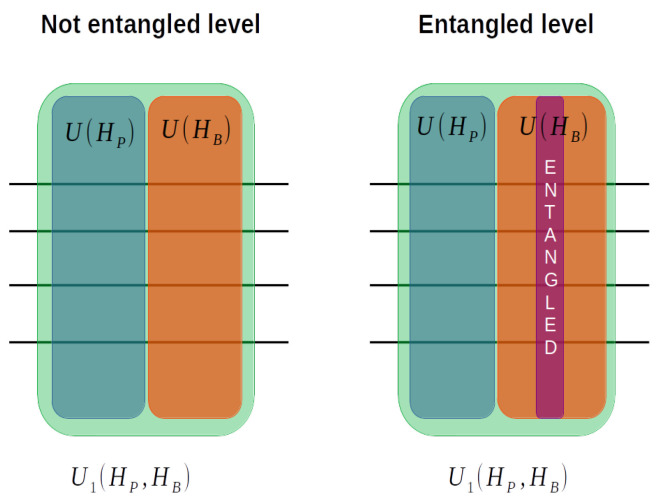
Non-entangled and entangled mixing operators.

**Figure 3 entropy-25-01499-f003:**
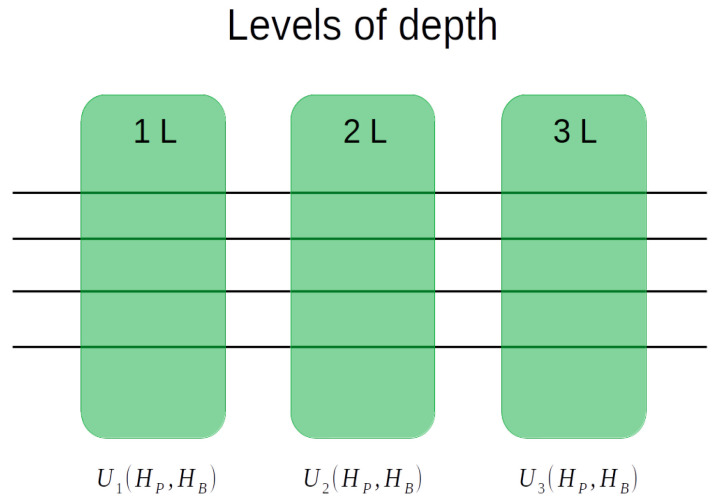
Levels of depth, with one pair of phase and mixing operator 1L, two pairs 2L, and three pairs 3L.

**Figure 4 entropy-25-01499-f004:**
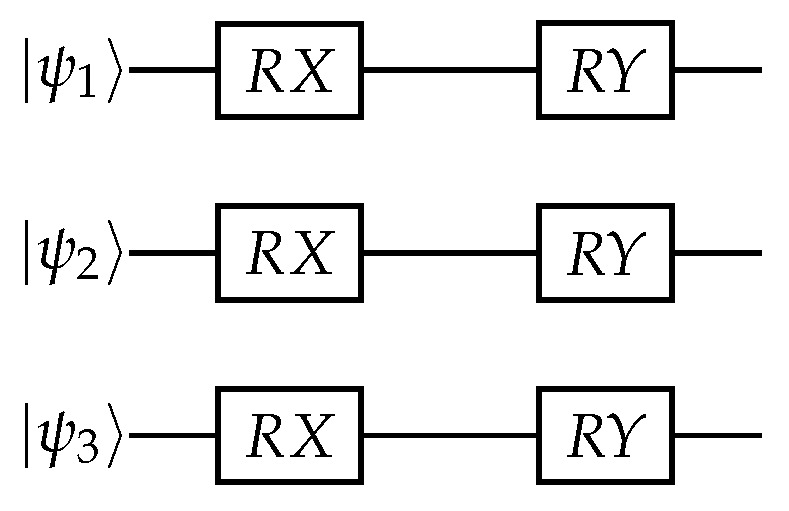
Individual rotations in mixing operators.

**Figure 5 entropy-25-01499-f005:**
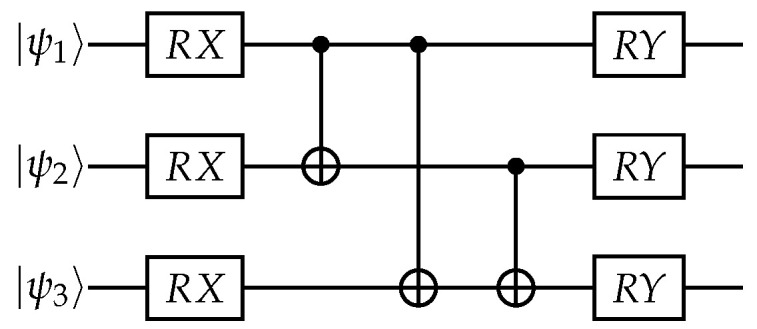
Entangled rotations in mixing operators.

**Figure 6 entropy-25-01499-f006:**
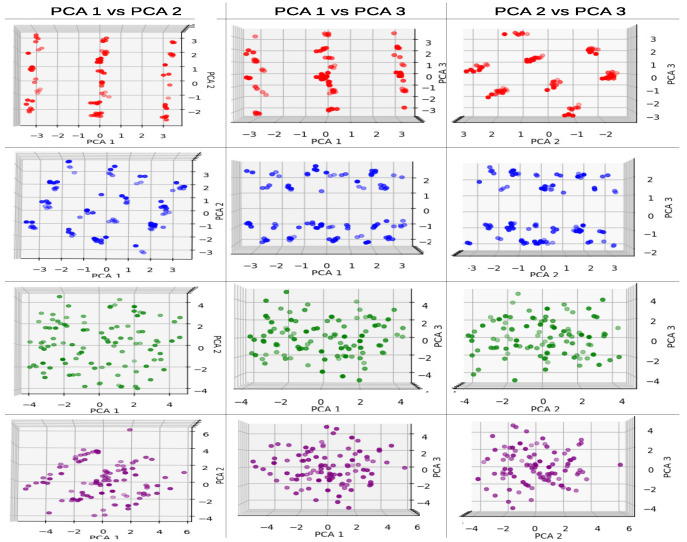
PCA individual graphs for 4n cyclic configuration max-cut problem solved using QAOA, first 3 components. Red corresponds to the 3p parameter 1L non-entangled, blue 3p parameter 1L entangled, green 6p parameter 2L non-entangled, and purple 6p parameter 2L entangled model.

**Figure 7 entropy-25-01499-f007:**
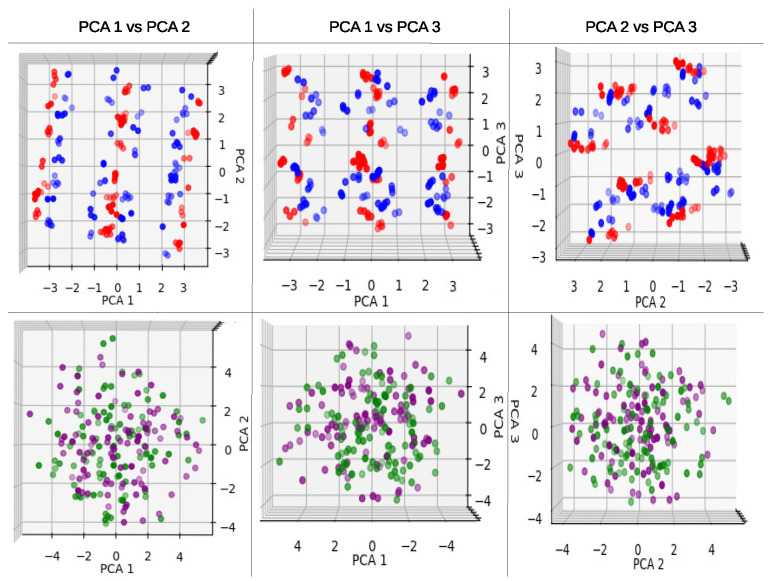
PCA pair graphs for 4n cyclic configuration max-cut problem solved using QAOA, first 3 components. Red corresponds to the 3p parameter 1L non-entangled, blue 3p parameter 1L entangled, green 6p parameter 2L non-entangled, and purple 6p parameter 2L entangled model.

**Figure 8 entropy-25-01499-f008:**
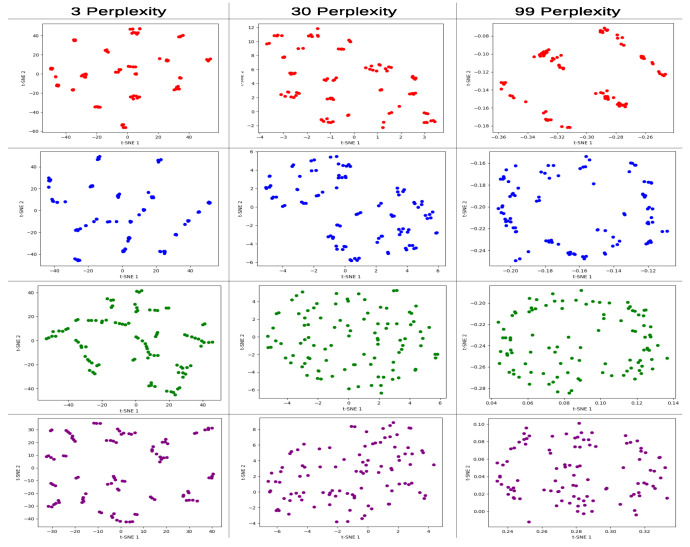
t-SNE individual graphs for 4n cyclic configuration max-cut problem solved using QAOA, with different perplexity values 3, 30, and 99. Red corresponds to the 3p parameter 1L non-entangled, blue 3p parameter 1L entangled, green 6p parameter 2L non-entangled, and purple 6p parameter 2L entangled model.

**Figure 9 entropy-25-01499-f009:**
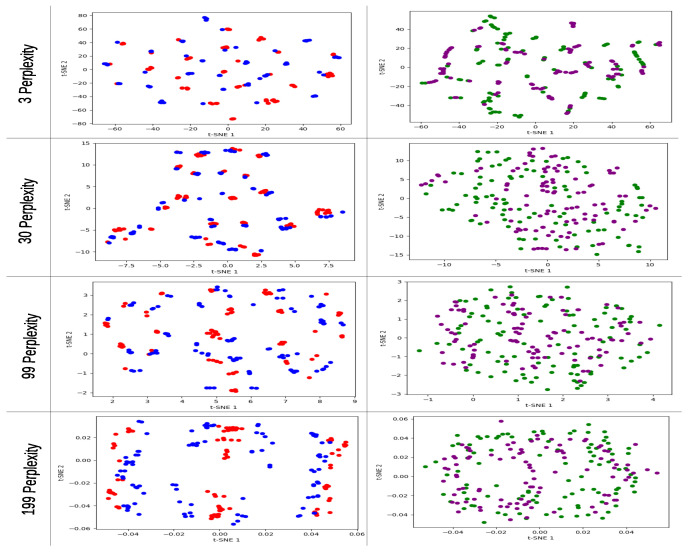
t-SNE pair graphs for 4n cyclic configuration max-cut problem solved using QAOA, with different perplexity values 3, 30, 99, and 199. Red corresponds to the 3p parameter 1L non-entangled, blue 3p parameter 1L entangled, green 6p parameter 2L non-entangled, and purple 6p parameter 2L entangled model.

**Table 1 entropy-25-01499-t001:** Individual PCA projections explained variance (4n cyclic) for the first 3 PCA components.

Parameters	PCA 1	PCA 2	PCA 3
3 p	0.44317179	0.29359184	0.26323637
3 p ent	0.4189022	0.30327991	0.2778179
6 p	0.23918661	0.20909359	0.16835319
6 p ent	0.2949565	0.22615379	0.1985824

**Table 2 entropy-25-01499-t002:** Pair PCA projections explained variance for the first 3 PCA components for the 4n cyclic max-cut problem.

Parameters	PCA 1	PCA 2	PCA 3
3 p	0.42701248	0.29829486	0.27469266
6 p	0.22597391	0.19226576	0.18187536

**Table 3 entropy-25-01499-t003:** Individual KL-Divergence for 4n cyclic max-cut problem with different numbers of perplexity, considering the 3p non-entangled, 3p entangled, 6p non-entangled, and 6p entangled models.

	KL-Divergence		
**Parameters**	**KL-D**	**KL-D**	**KL-D**
3 p	0.12658161	0.18968964	0.00003131
3 p ent	0.08551478	0.20780504	0.00004092
6 p	0.55863631	0.4951154	0.00003326
6 p ent	0.35603735	0.39002356	0.00004086

**Table 4 entropy-25-01499-t004:** Pair KL-Divergence for 4n cyclic max-cut problem with different numbers of perplexity, considering the 3p parameters (non-entangled and entangled) and 6p parameters (non-entangled and entangled) models.

	KL-Divergence	
**Parameter**	**3 per**	**30 per**
3 p	0.11827804	0.24608216
6 p	0.58829921	0.73377264
	99 per	199 per
3 p	0.17905515	0.00003183
6 p	0.33970055	0.00004709

## Data Availability

All the experiment data results can be found in https://github.com/BrianSarmina/QAOA_SHC-RR (accessed on 12 September 2023).

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
