# Peer review of "Principal Component Analysis and t-Distributed Stochastic Neighbor Embedding Analysis in the Study of Quantum Approximate Optimization Algorithm Entangled and Non-Entangled Mixing Operators"

_entropy, 2023, doi:10.3390/e25111499_

Round 1

Reviewer 1 Report

The paper applies Principal Component Analysis (PCA) and t-distributed Stochastic Neighbor Embedding (t-SNE) analysis to gain a deeper understanding of the behavior of mixing operators in the Quantum Approximate Optimization Algorithm (QAOA) at varying depths. 

They provide results, which I summarized as,

1.     Presenting difference between entangled and non-entangled models, which is quantified numerically through the explained variance for PCA and Kullback-Leibler divergence (after optimization) for t-SNE.

2.     Providing graphical representations in the mapping data of both PCA and t-SNE further emphasize the distinctions between entangled and non-entangled QAOA models.

Using the classical statistic methods to analysis is rare, and I am not familiar with t-SNE. From the part of PCA and what I can understand of t-SNE, I evaluate the paper with the following aspects.

Originality / Novelty

The understanding of the behavior of phase or mixing operators(we refer them as one body iteration and multi-body iteration terms), which are defined in this paper, is important.  And insightful study of different entangled quantum circuits used in QAOA is interesting.  

Significance of Content and Scientific Soundness

I am quite confused with the procedure and results.

The procedure and results are very vague.

First, they said, max-cut is solved, and they deal with optimized circuit with PCA or t-SNE. How to weigh the max-cut is solved. 

Moreover, as entangled circuit and non-entangled circuit would lead a different optimized cost, why we want to compare their parameter variance with different optimized results. 

Finally, once we have results from PCA and t-SNE, what can we learn from results? The motivation is also vague. 

The use of both PCA and t-SNE analysis techniques to the investigation, seems offering numerical and graphical evidence to support the observed differences. But what 

But they cannot convince me with this numerical case. 

Quality of Presentation

To be honest, I know PCA, but I still get lost in the introduction part when I read this classical method.  Furthermore, the abbreviation encountered for the first time should be in their complete form.

Interest to the readers

The study utilizes a dataset of parameters generated for max-cut problems, employing Stochastic Hill Climbing with Random Restarts as the optimization method within the QAOA framework. But the motivation is vague. 

From above reasons, I cannot recommend the publication of this article if my understanding is right and complete. 

Author Response

see attched

Reviewer 2 Report

The article can be improved in the following aspects:

- Provide more background about the problem to general audience in computer science and system. Not limited to Introduction, authors usually refer to new jargons without high-level explanation before, including "rotation gates in circuit", "deeper quantum circuits", "entangled operators", etc.

- Clarify the objective of the study. Authors mention the general objective in Introduction, as "to assess whether visualization strategies offer adequate information about the models". But in problem formulation, the focus is claimed on max-cut problem. Is max-cut a generic case study? Or is there any other strong correlation between those goals at different levels?

Mingled with many jargons, long sentences make it harder to clarify points. It would easier to follow using concise expressions.

Round 2

Reviewer 1 Report

Thank you for the detailed responses with modified part in the manuscripts, where I got your motivation and methodology. 

I think this version provides me more information and address a lot of my concerns.

However, some points are still unclear.

First, I know your primary focus is on pre-trained parameters of QAOA solutions.  You use dataset here to mean you use lots of pre-trained parameters set. How is the situation (or goodness to the answer) of these pre-trained parameters of QAOA solutions.  I think this should be addressed as this pre-training can be randomly initialized. 

Secondly, if my understanding is not wrong, for me, the most attraction thing in this work is to distinguish ansatz from the relationship between their local parameters. I advise this point can be emphasized. 
